# High entropy oxides for reversible energy storage

Abhishek Sarkar [1], Leonardo Velasco[1], Di Wang[1,2], Qingsong Wang[1], Gopichand Talasila[1], Lea de Biasi[1], Christian Kübel [1,2,3], Torsten Brezesinski [1], Subramshu S. Bhattacharya[4], Horst Hahn[1,3] & Ben Breitung [1,2]

In recent years, the concept of entropy stabilization of crystal structures in oxide systems has led to an increased research activity in the field of "high entropy oxides". These compounds comprise the incorporation of multiple metal cations into single-phase crystal structures and interactions among the various metal cations leading to interesting novel and unexpected properties. Here, we report on the reversible lithium storage properties of the high entropy oxides, the underlying mechanisms governing these properties, and the influence of entropy stabilization on the electrochemical behavior. It is found that the stabilization effect of entropy brings significant benefits for the storage capacity retention of high entropy oxides and greatly improves the cycling stability. Additionally, it is observed that the electrochemical behavior of the high entropy oxides depends on each of the metal cations present, thus providing the opportunity to tailor the electrochemical properties by simply changing the elemental composition.

---

[1] Institute of Nanotechnology, Karlsruhe Institute of Technology, Hermann-von-Helmholtz-Platz 1, 76344 Eggenstein-Leopoldshafen, Germany. [2] Karlsruhe Nano Micro Facility, Karlsruhe Institute of Technology, Hermann-von-Helmholtz-Platz 1, 76344 Eggenstein-Leopoldshafen, Germany. [3] Helmholtz Institute Ulm for Electrochemical Energy Storage, Helmholtzstraße 11, 89081 Ulm, Germany. [4] Department of Metallurgical and Materials Engineering, Nano Functional Materials Technology Centre (NFMTC), Indian Institute of Technology Madras, Chennai 600036, India. Correspondence and requests for materials should be addressed to H.H. (email: horst.hahn@kit.edu) or to B.B. (email: ben.breitung@kit.edu)

The demand for energy storage devices (batteries) for both stationary and mobile applications has increased rapidly during the past years and it is expected to continue to grow in the future. The most commonly used electrochemical energy storage devices are intercalation based Li-ion batteries, which exhibit very high efficiency and reversibility[1,2]. Nonetheless, other Li-storage schemes are being presently pursued especially conversion or alloying modification approaches since they hold the promise for achieving very high capacity storage systems. Unfortunately, many of these systems have been found to lack both good reversibility and efficiency[3,4].

Recently, a new class of oxide systems, also known as high entropy oxides (HEO), was formulated and reported with first demonstrations for transition-metal-based HEO (TM-HEO)[5–7], rare-earth-based HEO (RE-HEO)[8] and mixed HEO (TM-RE-HEO)[9]. HEO are based on a new, quite revolutionary concept of entropy stabilization, that is, to stabilize a certain crystal structure that can differ from the typical crystal structures of the constituent elements, thereby increasing the configurational entropy of the resulting compounds. This concept was first reported for metallic high entropy alloys (HEA). In recent years, the study of HEA has grown into an independent field of materials research, as evidenced by numerous publications[10].

Several reports on TM-HEO[5,6], RE-HEO[8], and mixed TM-RE-HEO[9] have demonstrated that high entropy stabilization in oxides with 5 or more cations in equiatomic concentrations leads to the formation of single-phase rock-salt, fluorite, or perovskite structures. These compounds often show interesting and unexpected properties, such as extraordinarily high room temperature Li-ion conductivities for solid state electrolytes in TM-HEO[5], very narrow and tailored band gaps in RE-HEO[8] and colossal dielectric constants in TM-HEO[11]. The main driver for the growing interest in HEO is the potential to obtain novel properties by exploiting the enormous number of possible elemental combinations[5,12]. The fast growth of the field of HEO is being facilitated by the availability of many synthesis and processing routes, which were shown to provide highly reproducible material systems[6,7].

As previously mentioned, one of the unexpected properties of the TM-HEO is the high Li-ion conductivity ($>10^{-3}$ S cm$^{-1}$), as reported by Bérardan et al.[5]. The possible insertion of Li-ions into a rock-salt structure opens several diffusion pathways for Li-ions through the crystal lattice, giving rise to the increased conductivity.

Here we present new results on the electrochemical properties of several TM-HEO, such as storage capacity and the cycling stability of HEO structures. The concept of high entropy crystal structure stabilization enables us to build conversion-based electrodes, which can be cycled over 500 times without significant capacity degradation. It is shown that the reduction of the entropy by removal of a single element leads to a completely different electrochemical behavior and cycling degradation. Additionally, unique possibilities to fine-tune the electrochemical performance of high-entropy materials is demonstrated, by making use of the different effects of each individual element on the electrochemical characteristics. Moreover, based on TEM investigations a possible reaction mechanism is proposed, which considers the entropy stabilization and the supporting rock-salt matrix structure during the entire conversion process.

## Results

### As-prepared TM-HEO characterization.
The morphology of the various TM-HEO powders, as reported in an earlier publication[6], comprises both hollow and filled spheres with sizes in the nano-to-micrometer range. The overview scanning electron microscopy (SEM) image in Figure 1a shows the variety of particle morphologies and sizes. A higher magnification view of the individual particles, both hollow and filled, can be seen in the inset. The crystallinity and the phase purity (rock-salt structure) of the TM-HEO (($Co_{0.2}Cu_{0.2}Mg_{0.2}Ni_{0.2}Zn_{0.2}$)O) were examined by means of powder X-ray diffraction (XRD), followed by Rietveld refinement analysis. More details about Rietveld refinement can be found in the supporting information (Supplementary Figure 1). The prepared TM-HEO was subsequently used as active material for application in Li-cells without any further heat treatment step.

The ($Co_{0.2}Cu_{0.2}Mg_{0.2}Ni_{0.2}Zn_{0.2}$)O electrodes were tested in secondary Li-based battery cells, using 63 wt% of the TM-HEO as active material and evaluated at different specific currents during cycling. Figure 1b, c depict representative data for this TM-HEO system. Galvanostatic cycling experiments were performed in the voltage range from 0.01 to 3 V with respect to Li$^+$/Li. The most probable reaction mechanisms are insertion of Li into the rock-salt HEO crystal structure, or a conversion reaction forming metals and Li$_2$O[3,13]. The most prominent materials for Li-insertion (intercalation)[2] are represented by layered structures like graphite, lithium cobalt oxide (LCO)[1], or lithium nickel cobalt manganese oxide (NCM) which are frequently used as electrode materials in Li-ion batteries, whereas insertion processes in rock-salt structures have been rarely reported[14]. The precondition for insertion in the TM-HEO would be that the Li-ion fits into the rock-salt lattice, where all octahedral positions are occupied by cations. As per Pauling's rule, the possibility of accommodating Li$^+$ in the tetrahedral position is energetically costly considering that the ratio of ionic radii (Li$^+$ to O$^{2-}$) is greater than 0.414 (the upper limit for accommodating an ion in the tetrahedral position)[15]. Additionally, this would lead to strong repulsive interactions between the cations in the lattice[15]. As known from the literature, transition metal oxides are prone to conversion reactions, i.e., during the charging-discharging cycle complete phase transformations may occur[14,16]. Consequently, it is reasonable to assume that reversible conversion reactions take place in TM-HEO, which either completely or partially reduce the metal ions upon lithiation.

The measured capacities at various specific currents of the TM-HEO are presented in Fig. 1b. Conversion electrodes often show substantial capacity degradation at high currents due to kinetic limitations of diffusion driven processes during de-/lithiation. For conversion type materials, the theoretical capacity is directly related to the amount of electrons transferred per formula unit (Supplementary Equation 1). The basic reaction for a binary metal oxide can be represented as $MO_x + 2x\text{Li} \rightarrow M + x\text{Li}_2O$. Because it is not known what reactions are occurring during the redox processes in the case of TM-HEO (additional processes might be alloying of Zn with Li, formation of intermetallic phases etc.), we cannot predict with high accuracy the theoretical capacity of our compounds. However, we believe that it is in the range of capacities reported for divalent oxide conversion materials (700–1000 mAh g$^{-1}$).

Although some conversion materials have been reported to show high specific capacities and high degrees of reversibility, most of them are tediously modified, e.g., regarding morphology and structure or they are coated with additional functional materials to enhance their electrochemical performance[17–19]. Usually, the particle size in conversion materials has a considerable influence on the electrochemical properties, since large size particles usually lead to low capacity as well as low rate capability and reversibility. Nevertheless, despite the large size particles present in the TM-HEO, the material shows high specific capacities even when applying high specific currents. Furthermore, it is able to fully recover and even increase the initial capacity after raising the current to 3 A g$^{-1}$ for 5 cycles,

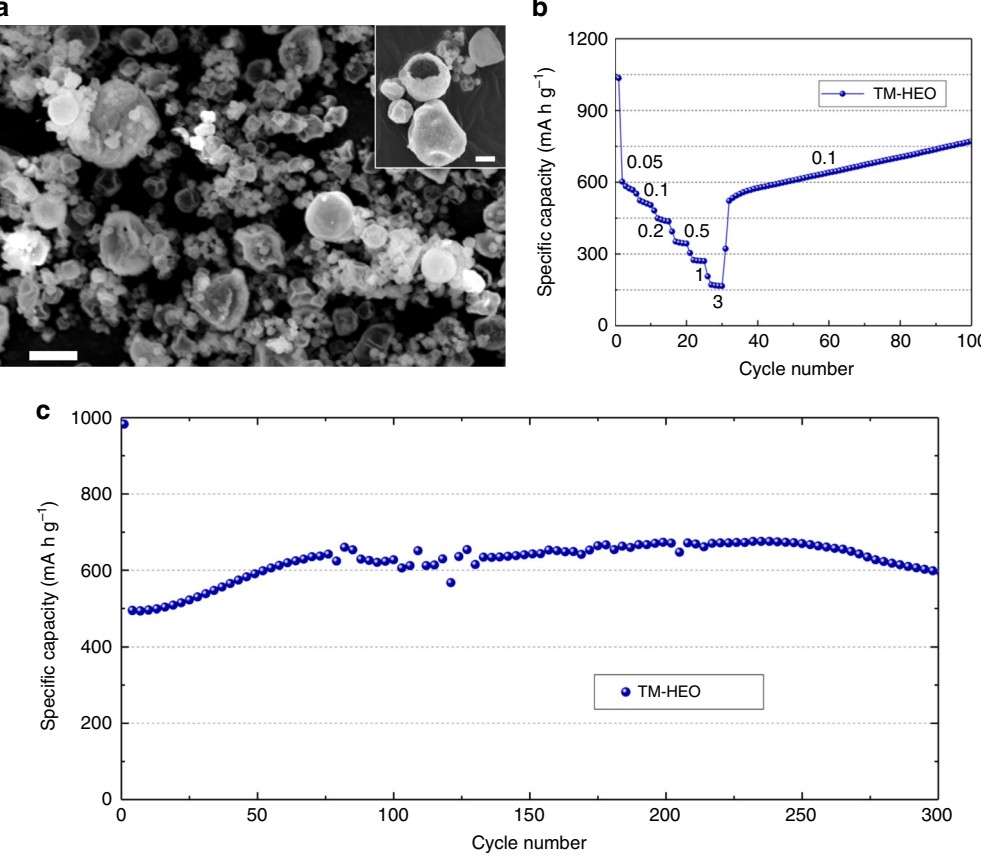

**Fig. 1** TM-HEO particles and their specific capacities. **a** SEM micrograph of the as-synthesized TM-HEO powder with typical particle sizes from the nano-to-micrometer range. The inset shows that the powder contains both hollow and filled spheres. The scalebars shown in **a** and the inset correspond to 2 μm and 1 μm, respectively. **b** Galvanostatic rate capability test (current values given in units of A g⁻¹). The capacity decreases as the specific current is increased stepwise up to 3 A g⁻¹, but it recovers at 0.1 A g⁻¹ and even increases after 100 cycles to 770 mAh g⁻¹. **c** Long-term cycling performance at 0.2 A g⁻¹

so that after 100 cycles a specific capacity of 770 mAh g⁻¹ is reached. Capacity increase over prolonged cycling is typical of conversion type materials and can be attributed to activation processes, occurring in electrodes with large particle sizes[14,17]. Figure 1c shows the cycling performance of the TM-HEO over 300 cycles at 200 mA g⁻¹ with two initial formation cycles at 50 mA g⁻¹. Even with the micrometer-sized particles, as shown in Fig. 1a, and without any optimization of the other components in the electrochemical cell (i.e., electrolyte, binder and electrode composition), the cells display good stability at high capacity values, especially when considering the conversion type of reaction involved. The aforementioned increase in capacity is also observed during the first 75 cycles. The initial discharge capacity amounts to 980 mAh g⁻¹, and after stabilization at the third cycle, the cell reaches a capacity of ~600 mAh g⁻¹, which even increases to around 650 mAh g⁻¹ after 70 cycles; the voltage profiles can be found in the supporting information (Supplementary Figure 2). The fluctuations in capacity seen between the 75th and 150th cycles were observed for several different cells. However, after around 150 cycles, the capacity was found to stabilize, therefore we attribute this behavior to structural changes associated with the active material. The conversion reactions of individual transition metal oxides (CuO, NiO, CoO etc.) with similar particle sizes and shapes show comparable initial specific capacities (500–700 mAh g⁻¹)[14,20,21], but they fall short when it comes to capacity retention and efficiency[22,23].

**Comparison between high entropy and medium entropy compounds**. The substantial number of cation metals in the TM-HEO makes it possible to remove specific elements and to then investigate the resulting change of the electrochemical behavior, thus allowing to assign certain electrochemical characteristics to specific elements. Additionally, the exclusion of one of the elements from a 5-cation system results in a significant decrease of the configurational entropy from ~1.61 R to ~1.39 R, which necessitates a post annealing treatment to obtain a single-phase oxide, as explained in the experimental section of the paper. To compare the 4-cation systems with the 5-cation system, the latter was subjected to the identical additional heat treatment as that used for the 4-cation oxides to facilitate comparison. In fact, in a long-term experiment (Fig. 2a), it was shown that, at a specific current of 200 mA g⁻¹, the 5-cation sintered system can be cycled over hundreds of cycles with specific capacities of up to 590 mAh g⁻¹. The typical electrochemical behavior of conversion materials discussed above (decaying/increasing capacity with cycling) is also apparent in this experiment. As anticipated, the first drop in capacity is more prominent since the calcined TM-HEO sample comprises even bigger particle sizes and agglomerates than the as-prepared TM-HEO (Supplementary Figure 3). The Coulombic efficiency in the stable region (cycle no. 60–400) stabilizes between 99.4 and 99.95%. Because of the heat treatment, slightly different capacity values for the 5-cation system were expected, as shown in Fig. 2a. According to the established nomenclature, 4-cation systems belong to the "medium entropy"

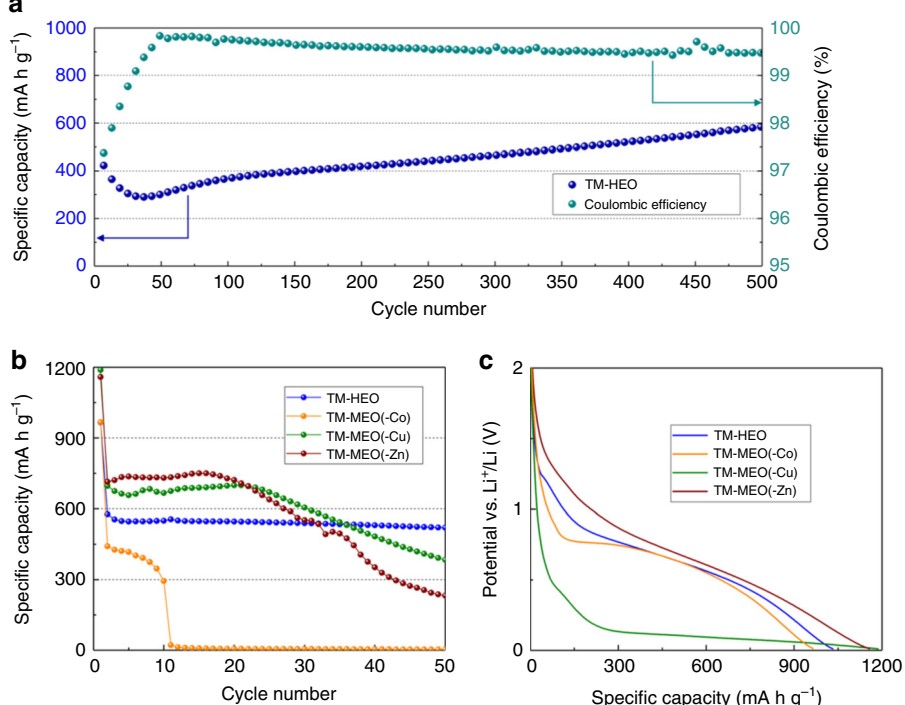

**Fig. 2** Specific capacities and lithiation profiles of TM-HEO and TM-MEO. Comparison of the different medium and high entropy oxides under investigation. **a** Long-term cycling stability of the calcined TM-HEO at 200 mA g$^{-1}$ together with the corresponding Coulombic efficiency. **b** It can be seen that the TM-HEO shows stable capacity retention at 50 mA g$^{-1}$, while the materials without Zn and Cu reveal severe capacity degradation. The material without Co fails completely after 10 cycles. **c** Discharge (lithiation) profiles of the first cycle for the different compounds. The material without Cu shows a significantly lower discharge potential and it might be interesting as anode material for primary batteries

| | Entropy | Removed element | Abbreviation |
|---|---|---|---|
| $(Co_{0.2}Cu_{0.2}Mg_{0.2}Ni_{0.2}Zn_{0.2})O$ | 1.61 R | – | TM-HEO |
| $(Co_{0.25}Cu_{0.25}Mg_{0.25}Ni_{0.25})O$ | 1.39 R | Zn | TM-MEO(-Zn) |
| $(Co_{0.25}Mg_{0.25}Ni_{0.25}Zn_{0.25})O$ | 1.39 R | Cu | TM-MEO(-Cu) |
| $(Cu_{0.25}Mg_{0.25}Ni_{0.25}Zn_{0.25})O$ | 1.39 R | Co | TM-MEO(-Co) |

**Table 1 Synthesized single-phase compounds used for electrochemical testing**

group, while 5-cation systems belong to the "high entropy" group[24]. It should be noted that all "medium entropy" oxides (MEO) showed a completely different and rather unstable electrochemical behavior when compared to the "high entropy" oxides (Fig. 2b). The electrochemical behavior of the "medium entropy" oxides, initially displaying high specific capacities, but then showing varying degrees of degradation with increasing numbers of cycles, compares well with that of established conversion materials with large particle sizes. By contrast, the 5-cation high entropy oxide displays a high capacity value that does not degrade with increasing cycle number. The capacity comparison between the 5-cation TM-HEO and the 4-cation oxides leads us to conclude that the "high entropy" material exhibits a novel and interesting electrochemical behavior, probably related to the entropy stabilization. Such an observation has not been previously reported. Figure 2b shows a comparison of the electrochemical performance of the 5-cation TM-HEO and three of the 4-cation systems, without Zn, Cu, or Co, respectively (see Table 1). Despite efforts to synthesize every possible MEO structure, both TM-MEO(-Mg) and TM-MEO(-Ni) (i.e., $(Co_{0.25}Cu_{0.25}Zn_{0.25}Ni_{0.25})O$ and $(Co_{0.25}Cu_{0.25}Mg_{0.25}Zn_{0.25})O$, respectively) could not be stabilized as single-phase compounds,

even at much higher temperatures. Therefore, we refrain from including them in the comparison of the entropy-stabilized oxides, since we believe that simple mixtures of different compounds are not comparable to the single-phase materials. Nevertheless, the electrochemical and XRD characterization results of the multiphase compounds are depicted separately in Supplementary Figure 4.

The configurational entropy was calculated using the following formula:[24]

$$S_{config} = -R \left[ \left( \sum_{i=1}^{N} x_i \ln x_i \right)_{cation-site} + \left( \sum_{j=1}^{N} x_j \ln x_j \right)_{anion-site} \right]$$ (1)

where $x_i$ and $x_j$ represent the mole fractions of ions present in the cation- and anion-site, respectively. The contribution of the anion-site is expected to have a minor influence on $S_{config}$, given that only one anion is present. More details about the entropy calculation for the TM-HEO and one of the TM-MEO materials are given in Supplementary Equation 2.

Although every composition was synthesized as single-phase rock-salt structure (the XRD patterns of all the compounds can be found in Supplementary Figure 5), significant differences in the electrochemical behavior between the TM-HEO and the "medium entropy" materials are clearly evident. The individual compounds are cycled at 50 mA g$^{-1}$ in a potential range between 0.01 and 3 V with respect to Li$^+$/Li. Figure 2b shows the TM-HEO with a specific capacity of 555 mAh g$^{-1}$ after the initial formation cycles, which decreases to 520 mAh g$^{-1}$ after 50 cycles. The removal of Co (TM-MEO(-Co)) leads to a complete failure of the cell after approx. 10 cycles, with no signs of recovery during the subsequent cycles (Fig. 2b). This seems to imply that Co can

be considered as a critical and necessary element for TM-HEO to have high specific capacity and good cycling stability, whereas the removal of Zn (TM-MEO(-Zn)) and Cu (TM-MEO(-Cu)) does not impede the overall reversibility. Nevertheless, the cells deliver lower capacities after 30–35 cycles, even though the initial capacity of both oxides is higher than that of the 5-cation system, but with a rapid drop after only a few cycles. Similar electrochemical behavior has been reported for large particle size conversion materials and has often been explained by side reactions, which occur during the redox processes, such as particle fracture/pulverization and increased solid-electrolyte interphase formation[2]. Another aspect discussed in the literature is the potential improvement in conductivity when metallic species like Cu dendrites are present[25], which might lead to better cycling stability, too. To examine whether Cu dendrites are formed during the conversion process, energy-dispersive X-ray spectroscopy (EDX) measurements were conducted on electrodes after 100 cycles. However, the data show no indication of any Cu aggregation in the materials (Supplementary Figure 6)[26].

Therefore, the 4-cation systems can serve as reference materials when the entropy stabilization is not as large as in the HEO. Such effects were not observed during TM-HEO cycling. SEM examination of the cycled materials revealed fully intact spheres, embedded in the carbon/polymer matrix of the electrode (Supplementary Figure 7).

The above results demonstrate that an increase in concentration (25 at% in 4-cation systems vs. 20 at% in 5-cation systems) of constituent elements, which are known for their electrochemical activity (e.g., Co), leads to the expected increase in capacity, but the 4-cation systems do not exhibit the stability of the 5-cation system. Another interesting effect due to removal of Cu from the TM-HEO has been noted. As seen in Fig. 2c, the initial (average) discharge potential is significantly decreased. The large capacity of this compound at the low potential could be an interesting option for primary anode applications (around 800 mAh g$^{-1}$ in the 0.03–0.13 V range).

The cyclic voltammograms (CVs) are depicted for all the systems in Supplementary Figure 8. The CV curves look very different for the different materials, which is another clear indication that the reaction mechanism and the electrochemical behavior can be tailored by changing the elemental composition. Removal of Zn from the TM-HEO causes a completely different electrochemical behavior during the oxidation of the compound. The CVs and differential capacity plots establish that the absence of Zn leads to a two-step oxidation process, rather than a single one as for all the other samples (Supplementary Figure 9). These two oxidation peaks, centered around 1.7 V and 2.2 V, resemble the formation of individual NiO and CoO[17,27]. This suggest that, in the case of TM-MEO(-Zn), the parent rock-salt structure is apparently not regained upon Li extraction, but instead separates into different phases.

Despite potential alloying of elemental Zn with Li[28], we did not find any signs of this reaction. Usually, the formation of ZnLi occurs at a low potential of around 0.2 V[28] and would lead to an additional gain in capacity of ~40 mAh g$^{-1}$ for the TM-HEO[29]. Although the CV curves (Supplementary Figure 8) for TM-MEO (-Zn) and the Zn-containing materials do not show significant differences, alloying of Zn with Li cannot be ruled out completely, especially for TM-MEO(-Cu), where the vast majority of Li uptake occurs at low potential. However, even if this reaction takes place in the TM-HEO, the capacity gain will only account for ~6.7% of the total capacity.

These examples for TM-MEO(-Cu), TM-MEO(-Zn), and TM-MEO(-Co) demonstrate that even the removal of a single element leads to significant changes in the electrochemical properties. However, the changes are different for each element removed. On the other hand, the addition of other cations into the single-phase structure (e.g., Fe, Mn, Cr, V) could open new possibilities for developing a modular method to tailor electrodes suited to particular needs. The opportunities to tailor the electrochemical properties in such a flexible way (countless cation combinations are possible), as offered by HEO, are unique compared to conventional conversion or intercalation electrodes. In the following sections of the paper, a possible interpretation, related to the entropy stabilization of the crystal structure, is presented.

**Structural investigations and influence on the reaction mechanism.** To better understand the reactions and the underlying mechanism of reversible lithium storage, a comprehensive characterization using XRD and transmission electron microscopy (TEM) was performed. Two possible pathways exist for a reversible conversion reaction: (1) the initial TM-HEO phase/structure is being (re)transformed with each lithiation/delithiation cycle or (2) only a few distinct elements are participating in the phase transformation, while the others are keeping the rock-salt structure intact. A full transformation of the stable single-phase TM-HEO with a rock-salt structure is not expected, considering the high synthesis temperature of above 1000 °C in an oxygen atmosphere.

To gather more information on the lithiation mechanisms, operando XRD was conducted during the first two cycles of the cell. These measurements were correlated with high-resolution (HR) TEM and selected area electron diffraction (SAED) studies. The operando XRD measurements were performed on a TM-HEO electrode in transmission geometry. Figure 3 depicts the initial discharge/charge cycle. The intense reflections appearing over the whole cycle originate from the Cu current collector. This is clearly inferred, since the reflections of the initial rock-salt structure vanish during the lithiation and do not reappear after delithiation. This behavior is typical of conversion materials and it is a result of the formation of small crystallites, which have sizes below the detection threshold of XRD[13,14]. Nevertheless, SAED measurements conducted on the as-prepared, lithiated and delithiated particles show that the reflections, originating from the rock-salt structure, corresponding to the (111) and (200) plane do not disappear, even when the sample is fully lithiated. The preservation of the rock-salt structure (if only partially) over the entire redox process, is identified as a likely reason for the observed stable cycling behavior. No reflections of other crystal structures were detected based on the XRD patterns. The presence of completely unreacted TM-HEO regions, which would appear as spots or sharper diffraction rings in the SAED in the lithiated and cycled state, can be ruled out. From the combined structural characterization, it becomes evident that the entire volume of the TM-HEO is modified, giving rise to a reduction of the size of the coherently scattering regions and their chemical composition initiated by the exchange of Li ions. As will be described in the model below, the diffuse SAED rings, observed after the first cycle, thus belong to the material, which participated in the conversion reaction as host matrix.

TEM micrographs of the as-prepared TM-HEO (Fig. 4a) show a high degree of crystallinity and reveal the long-range order, as also evident from the XRD pattern in Supplementary Figure 1. TEM investigations of cycled TM-HEO confirm the presence of small crystalline regions, which still exhibit a long-range order that is visible in the diffraction rings. However, grain boundaries and defects reduce the size of the uniform structures below the coherence length for the XRD and, thus, these structures are not visible in the operando XRD patterns[14,18].

In a typical conversion reaction, metal oxides are being reduced to zero-valent metals and lithium oxide with lithiation. Since,

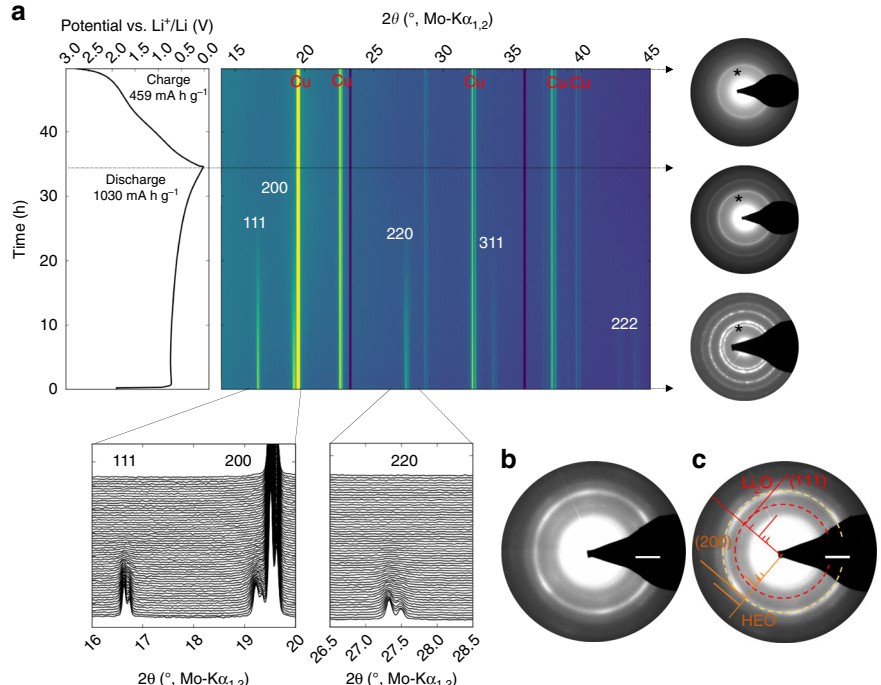

**Fig. 3** Operando XRD on TM-HEO. **a** XRD results obtained during the first full lithiation/delithiation cycle. The black lines with arrows indicate the as-prepared, fully lithiated and fully delithiated states as a function of time, with the corresponding potential curve shown on the left side. The XRD reflections fade with lithiation over the first few hours and, after the first full cycle, the reflections are not visible anymore. In the SAED patterns on the right side, crystallites, much smaller than the starting size, are still visible due to the shorter wavelength of the electrons. The asterisk indicates the (200) lattice plane of the rock-salt structure, which is maintained during the entire process. The presence of a very faint SAED ring in the lithiated samples in **b** and **c** (d-spacing 2.68 Å, scalebars 2 1/nm) could be attributed to the (111) diffraction of $Li_2O$. From the combined XRD and TEM results, it can be concluded that the rock-salt structure is preserved during cycling and serves as a host matrix for the redox processes. Unreacted TM-HEO can be ruled out since otherwise, the corresponding diffraction spots should be visible in the SAED and the XRD reflections should not disappear completely. Supplementary Figure 10 shows an SAED pattern obtained on an electrode after 10 full cycles, still showing the rock-salt structure. The voltage profile seen for the operando XRD measurement is plotted versus the dis/charge time. The first cycle discharge capacity amounts to around 1030 mAh g$^{-1}$, which corresponds to an uptake of roughly 2 Li per formula unit. The initial charge capacity is lower, being 459 mAh g$^{-1}$

these compounds are likely to have different crystal structures, compared to the as-prepared rock-salt structure of TM-HEO, a completely disordered structure after the delithiation cycle or separation into distinct elements could be expected. However, a pseudo long-range "ordered" structure, containing many defects, is observed (Fig. 4b, c).The lattice spacings, corresponding to the (111) and (200) planes of the rock-salt structure, are still clearly visible in the high-resolution image (Fig. 4b) and in its fast Fourier transform (FFT) (Fig. 4c). The experimental observations indicate that, even in the lithiated state, the rock-salt lattice is preserved and serves as a host structure for the conversion reactions. The cations involved in the conversion reactions can diffuse back during the lithiation and delithiation cycles. For the conversion reactions to occur, it is necessary that, during the lithiation process, some of the cations are reduced to the metallic state. However, it appears that these reduced cations remain "trapped" inside the crystal structure of the TM-HEO. It is suggested that the rock-salt structure is preserved by the remaining unreduced cations, which facilitates the re-occupation of the previously reduced cations to the original sites of the HEO lattice during the subsequent oxidation reaction. The observed defect structure is most likely the result of stresses in the crystal lattice (due to the conversion reactions of the participating metal elements), which leads to the size reduction of the crystallites and their disappearance in the diffraction signals.

To support the above hypothesis, the products of the conversion reaction ($Li_2O$ and metallic species) were analyzed

via TEM. Interestingly, no metallic phases were found, which seems to be the consequence of the aforementioned reaction mechanism (i.e., reduction inside the rock-salt matrix). Although $Li_2O$ is known to be highly sensitive to the electron beam, the presence of $Li_2O$ could be rationalized from the samples after the first and second lithiation cycles (see SAED patterns in Fig. 3a, b). The d-spacing of 2.68 Å could be attributed to the (111) reflection of $Li_2O$. Because $Li_2O$ is only forming due to the conversion reaction, this can be seen as an indirect support of the proposed reaction scheme.

Additionally, to support the hypothesis of a fully recovered TM-HEO rock-salt structure, spatially resolved EDX spectroscopy measurements were performed to rule out any segregation of elements and/or possible changes in the elemental composition of the materials during cycling. Figure 4d depicts scanning TEM (STEM)-EDX maps of the TM-HEO after the initial cycle. As shown in Supplementary Figure 11, EDX of the as-prepared material does not show any qualitative differences compared to the cycled TM-HEO.

The stability of the "high entropy" materials, when compared to the "medium entropy" compounds, can be explained in terms of the higher absolute value of configurational entropy, which decreases the Gibbs free energy in the TM-HEO structure. While the conversion of the lithiated TM-HEO back to the original rock-salt structure will be favored by the reduced Gibbs free energy, the TM-MEO with one element removed are not sufficiently entropy stabilized to ensure full transformation. The

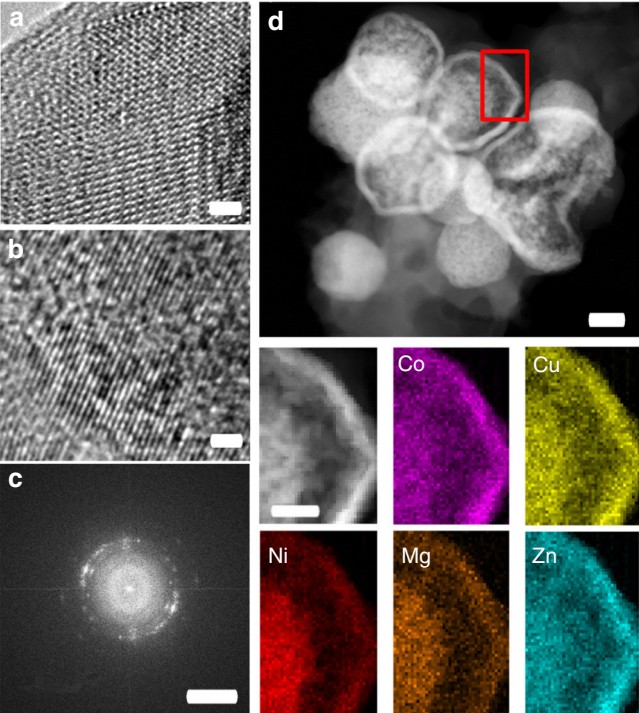

**Fig. 4** HRTEM and EDX analysis of the active material. **a**, **b** HRTEM images of the as-prepared and cycled TM-HEO, respectively. The crystallites in **b** are substantially smaller and do not show order over longer distances. The lattice fringes in **b** are not completely straight, but show a small divergence from the original axis. Nevertheless, the small regions with dimensions on the order of a few nanometers exhibit lattice fringes oriented in a specific direction, likely due to the former long-range order, which is partially lost during the conversion reaction. **c** FFT of a HRTEM image of a few crystallites. The arcs in the FFT correspond to the cubic rock-salt structure, but they indicate a certain texture. The lattice fringes are discontinuous between different crystallites; however, the misorientation is small, caused probably by large amounts of defects or low angle boundaries. **d** STEM image showing the spheres consist of small particles and the respective elemental maps of the area indicated by the red rectangle. No apparent segregation occurs at the length scale of aggregates of nanoparticles. The scalebars in **a** and **b** correspond to 1 nm, in **c** to 5 1/ nm, in **d** to 200 nm and for the compared EDX analysis to 60 nm

argument of higher stability is also evident from the fact that significantly longer heating times during or after synthesis are required for stable medium entropy oxides compared to the high entropy oxides. Supporting this assumption is the altered oxidation process presented in Supplementary Figure 9 for TM-MEO(-Zn). This material undergoes similar conversion reactions, exhibiting the well-known drawbacks of limited capacity retention and low cycling stability. By contrast, the TM-HEO shows a much more stable electrochemical performance, likely associated with entropy stabilization. Additional support for this hypothesis is also provided by the details of the Coulombic efficiency during cycling. While the TM-HEO provides values in the range between 98.5 and 99.5%, the TM-MEO(-Zn) exhibits Coulombic efficiencies substantially lower (85–95%) over the first 50 cycles (Fig. 5a). This difference alone is already indicative of important side reactions, reducing the efficiency when the entropy stabilization of the active material is not sufficient. For comparison of the "high entropy materials", "medium entropy materials" and "multiphase materials", the Coulombic efficiencies over the first 50 cycles are depicted in Fig. 5a. As is seen, only the "high entropy

material" shows stable behavior and can be cycled for hundreds of times. Even higher loading of TM-HEO cells reveals Coulombic efficiencies of >98.5% (Fig. 5b). In addition, Fig. 5b shows results from the respective rate performance test, where the areal capacity after 30 cycles amounts to 1.3 mAh cm$^{-2}$. Despite the fact that the loading was increased by a factor of ~5 (0.5 vs. 2.3 mg cm$^{-2}$), the specific capacity decreased only slightly (see also electrochemical performance of an electrode containing 80 wt% of TM-HEO in Supplementary Figure 12).

**Proposed conversion-based mechanism**. A schematic of the proposed reaction mechanism is illustrated in Fig. 6. Within the large as-prepared TM-HEO particles, which are poly-/nanocrystalline, lithiation induces conversion reactions of some of the cations (e.g., Co$^{2+}$, Cu$^{2+}$) while the other cations stabilize the rock-salt structure, thereby acting as a kind of matrix. We believe that the instance that single-phase compounds without Mg$^{2+}$ or Ni$^{2+}$ apparently cannot be synthesized is an indication of the stabilizing role of those ions to keep the structure intact during the redox processes. The pivotal role of Mg$^{2+}$ in keeping the HEO structure intact can be explained by the fact that Mg is inactive in the given potential range. The conversion reactions occur on much smaller length scales, on the order of several nm, while the crystallite sizes in the as-prepared powder are much larger. A phase separation of the elements, forming known conversion materials such as ZnCo$_2$O$_4$, can be ruled out, since the corresponding crystal structures could not be identified, even in the SAED measurements.

The preservation of the rock-salt structure even during lithiation is completely new and is unexpected from the traditional conversion reaction point of view. The fact that TEM and SAED results reveal highly disordered defect-rich rock-salt regions, without any second phase being present, is in good agreement with the proposed mechanism (Fig. 6). The re-incorporation of the metal ions into the TM-HEO rock-salt structure at room temperature upon electrochemical cycling, facilitated by high entropy stabilization, is expected to add a new dimension to the traditional conversion based lithiation mechanism.

## Discussion

In this study, to our knowledge for the first time, it is shown that high entropy oxides are very promising materials for reversible electrochemical energy storage. The variation of the composition of the oxides allows tailoring the Li-storage properties of the active material. The incorporation of different elements into HEO offers a modular approach for the systematic design of the electrode material. Additionally, it is shown that entropy-stabilized oxides have high capacity retention and exhibit a de-/lithiation behavior, which is drastically different from classical conversion materials. The new effect is attributed to configurational entropy stabilization of the lattice, which conserves the original rock-salt structure while serving as a permanent host matrix for the conversion cycles. Based on these—necessarily limited—first, but promising results, further investigations toward high entropy oxide electrode materials should be pursued to explore their full potential for energy storage applications.

## Methods

**Synthesis**. A versatile synthesis technique proven to yield highly crystalline HEO is the Nebulized Spray Pyrolysis (NSP) method, the details of which have been reported elsewhere[6]. In the NSP method, a solution, containing metal salts, is sprayed as a mist and then transported by means of a carrier gas (typically containing oxygen) into the hot zone of a tubular furnace. At the elevated temperature, the precursor solution transforms into the desired crystalline oxide. As the composition of the final product, a nanocrystalline oxide powder, is determined by the composition of the precursor solution, the process is highly reproducible and

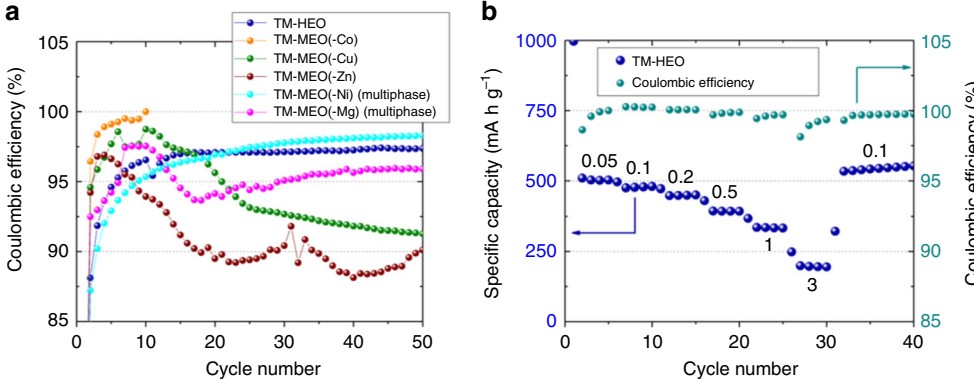

**Fig. 5** Coulombic efficiencies of the TM-MEO and TM-HEO compounds and rate test of a higher-loaded TM-HEO electrode. **a** Coulombic efficiency vs. cycle number for all the tested electrode materials at 50 mA g$^{-1}$. Only TM-HEO reveals stable cycling behavior. The multiphase material without Ni exhibits comparable efficiencies, but with much lower specific capacities (below 250 mAh g$^{-1}$ after 40 cycles) (Supplementary Figure 4). **b** Rate performance test performed on a TM-HEO cell with an overall higher loading (2.3 mg$_{\text{TM-HEO}}$ cm$^{-2}$). The current values in **b** are given in units of A g$^{-1}$

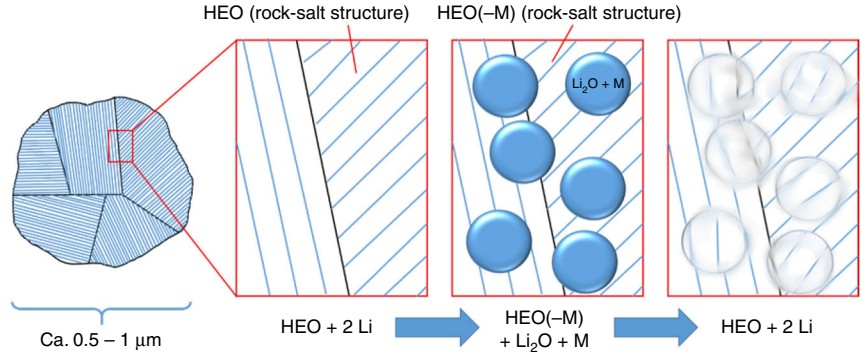

**Fig. 6** Schematics of the proposed de-/lithiation mechanism during the conversion reaction of TM-HEO. M in the figure stands only for the cations Co, Cu, Zn, and Ni, since Mg is electrochemically inactive in the potential range applied here. The as-prepared TM-HEO is made of poly-/nanocrystallites, exhibiting an ordered structure, as evidenced by SAED, TEM, and XRD. During the lithiation, some of the divalent metals of the TM-HEO react with Li to form nano-Li$_2$O and nano-M nuclei via a conversion reaction. The SAED measurements clearly show that the rock-salt structure is preserved in this state. The nanosized nuclei grow inside the rock-salt host structure, causing stresses to build up, thus resulting in the introduction of defects. Consequently, the reflections in the XRD pattern disappear—the nuclei "destroy" the long-range order; nevertheless, the participating ions remain "trapped" inside the host matrix and can easily diffuse back into the crystal structure in the subsequent oxidation process. Hence, the parent HEO structure is restored after delithiation

provides quantities on the scale of 1–2 g h$^{-1}$ for a laboratory reactor. In the present study, the respective transition metal ions were dissolved in an aqueous-based solvent using the corresponding nitrates ((Co(NO$_3$)$_2$·6H$_2$O (Sigma Aldrich, 99.9%), Cu(NO$_3$)$_2$·2.5H$_2$O (Sigma Aldrich, 99.9%), Mg(NO$_3$)$_2$·6H$_2$O (Sigma Aldrich, 99.9%), Ni(NO$_3$)$_2$·6H$_2$O (Sigma Aldrich, 99.9%), and Zn(NO$_3$)$_2$·6H$_2$O (Alfa Aesar, 99.9%)). The particles are formed in the gas phase of the hot-wall reactor, operated at 1150 °C. For the 5-cation system, a single-phase structure was obtained directly during the synthesis process. By contrast, the 4-cation systems (see Table 1) required a post annealing treatment (1 h at 1000 °C under ambient atmosphere) to form a single-phase compound. This annealing step was also applied to the 5-cation system, to make sure that all the compounds had the same synthesis/heat treatment history to facilitate comparison of the electrochemical properties among the different systems.

**Electrode processing**. TM-HEO electrodes were prepared by casting a water slurry containing 63 wt% TM-HEO, 22 wt% Super C65 carbon black additive (Timcal) and 15 wt% Selvol 425 poly(vinyl alcohol) (Sekisui)[30] onto Cu foil (Gould Electronics). The resulting electrode tapes were dried in vacuum at 80 °C for 6 h. The typical areal loading of active material was 0.5–1 mg cm$^{-2}$ unless mentioned otherwise (2.3 mg cm$^{-2}$ for high-loading electrodes). Coin-type cells with 600 µm-thick Li metal foil (Albemarle Germany GmbH) and glass microfiber separator (Whatman, GF/A; GE Healthcare Life Sciences) were assembled inside an argon-filled glovebox ([O$_2$] < 0.5 ppm, [H$_2$O] < 0.5 ppm). The electrolyte used was 1 M LiPF$_6$ in a 3:7 weight mixture of ethylene carbonate and either dimethyl carbonate

or ethyl methyl carbonate (BASF SE). All capacities given are related to the mass of the active material.

**Characterization**. Operando XRD was performed using a high-intensity laboratory Mo-Kα$_{1,2}$ diffractometer, optimized for battery research. Details of the setup as well as a description of calibration procedures can be found elsewhere[31–33]. Powder XRD patterns were recorded using a Bruker D8 Advance diffractometer with a Cu-Kα radiation source and a LYNXEYE detector having a fixed divergence slit (0.3°). For operando XRD, coin cells, equipped with polyimide windows, were assembled inside a glovebox by stacking electrode (Ø 12 mm), glass fiber separator (Ø 17 mm) and Li metal foil anode (Ø 16 mm) and using 150 µl of electrolyte[33]. During the operando XRD experiment, the cell was cycled at 100 mA g$^{-1}$ in the voltage range between 0.01 and 3 V. 2D diffraction patterns were collected in transmission geometry with an exposure time of 300 s. The intensities of two consecutive patterns were added up and then integrated to obtain 1D data, resulting in a total time resolution of 600 s. Transmission electron microscopy (TEM) experiments were conducted on powder samples dispersed onto a carbon-coated gold grid. The samples were loaded onto a Gatan TEM vacuum transfer holder inside a glovebox and transferred to the TEM without exposure to air. The TEM samples were examined using a Titan 80–300 electron microscope (FEI), equipped with a CEOS image spherical aberration corrector, high angle annular dark field (HAADF) scanning transmission electron microscopy (STEM) detector (Fischione model 3000) and a Tridiem Gatan image filter (GIF). The microscope was operated at an accelerating voltage of 300 kV. SEM was performed on a ZEISS Gemini Leo 1530. Galvanostatic charge/discharge measurements were performed at room

temperature and at various specific currents of 50–3000 mA g$^{-1}$ in the voltage range between 0.01 and 3.0 V vs. Li$^+$/Li using a MACCOR battery cycler.

**Data availability**. The data used for this study are available from the corresponding authors upon request.

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

## Acknowledgements

One of the authors (Q.W.) acknowledges financial support by EnABLES. This project has received funding from the European Union's Horizon 2020 research and innovation program under Grant Agreement No 730957. H.H. and A.S. acknowledge financial support from the Helmholtz Association and the Deutsche Forschungsgemeinschaft (DFG) project HA/1344/43-1. We acknowledge support by Deutsche Forschungsgemeinschaft and Open Access Publishing Fund of Karlsruhe Institute of Technology.

## Author contributions

A.S. synthesized the materials, fabricated the electrodes and performed the electrochemical experiments and analysis. L.V. synthesized the materials and analyzed HRTEM micrographs. D.W. and C.K. conducted and analyzed HRTEM and EDX measurements. G.T. supported the synthetic efforts. L.d.B. performed and analyzed operando XRD measurements. Q.W. prepared the electrodes with higher loading and helped with CV analysis. T.B. supervised the XRD measurements and co-wrote the manuscript. S.B. developed the material synthesis and supported the synthesis efforts. H.H. and B.B. supervised the synthesis and experiments, directed the project and co-wrote the manuscript.

## Additional information

**Competing interests:** The authors declare no competing interests.

