## [Peer Review File · Nature Communications]

Reviewers' comments:

Reviewer #1 (Remarks to the Author):

The authors prepared a five element oxide crystallizing in the NaCl structure and investigated the electrochemical behavior as electrode material for Li based batteries. The samples were characterized by TEM based methods and the electrochemical behavior is compared to oxides consisting of four cations. The better electrochemical behavior of the five element material is traced back to entropy stabilization. While investigation of such oxides for LIB applications is interesting the authors do not provide much insight into the reaction mechanisms occurring during Li uptake and release. Operando XRD demonstrates that the NaCl structure is destroyed on the long range scale. SAED shows reflections of very small and may be highly defective NaCl-type crystalline regions. I'll suggest addressing the points raised below and resubmit the manuscript. If Li enters the NaCl structure only tetrahedral sites are available. But what is the Li-M distance? I'll assume much too short and in my opinion intercalation can be ruled out.

The authors mention that the experimental capacities match with those calculated but do not present formal reactions with accompanying capacities. The authors did not mention that Zn^{2+} can be alloyed with Li often leading to higher capacity than expected for reduction of Zn^{2+} to Zn. It is well documented that Mg^{2+} is inactive in the potential window applied here. Hence, Fig. 5 is incorrect.

I'm wondering why no four element oxide was tested which does not contain Mg^{2+} .

The authors used a relatively low loading of 63%. Why? Was the loading identical for the four metal oxides? What was the loading per cm^2 ?

Figure 3: It would be much better showing the amount of Li consumed during discharge/charge than the time.

It was reported in literature that Cu^{2+} is reduced to elemental Cu during discharge and Cu dendrites are formed which increase conductivity of the electrode material thus improving cycling stability. Is this a possible scenario for the Cu containing samples?

For all samples a strong capacity loss is observed in the first cycle. The authors should make a comment on this and should estimate the amount of Li lost in the first cycle. If the loss of Li cannot be avoided (only Li for SEI formation is beneficial) conversion materials are not really relevant for industrial applications.

Bragg reflections are not signals!

Reviewer #2 (Remarks to the Author):

High Entropy Oxides for Reversible Energy Storage

This manuscript reports the application of high entropy oxide as a stable anode material in the lithium-ion battery. Authors have shown that entropy stabilization is a critical factor for designing a multicomponent electrode material. Overall findings of the manuscript related to the high entropy anode materials are interesting but parts of the manuscript, especially its electrochemical reaction mechanism does not appear to be well justified. The manuscript can be accepted for publication if authors could justify the following queries.

1- In this study $(Co_{0.2}Cu_{0.2}Mg_{0.2}Ni_{0.2}Zn_{0.2})O_2$, containing 5 metals was synthesized and it is reported that this material shows better electrochemical stability as compared to its 4 metal derivative oxides, for example $(Co_{0.25}Cu_{0.25}Mg_{0.25}Ni_{0.25})O_2$, $(Co_{0.25}Mg_{0.25}Ni_{0.25}Zn)O_2$ and so on. When comparing the 5 metal high entropy oxide with 4 metal low entropy oxide why electrochemical performance of compositions like $(Co_{0.25}Cu_{0.25}Ni_{0.25}Zn_{0.25})O_2$ and $(Co_{0.25}Cu_{0.25}Mg_{0.25}Zn_{0.25})O_2$ are not compared? Authors should report the electrochemical performance of these compositions as well to further clarify the effect of entropy on the stabilization of this anode material.

2- Table 1 reports the entropy values of the selected compositions of anode materials but method of entropy calculation is not mentioned anywhere in the manuscript. Authors should write the method of entropy calculation or they should point some suitable reference to justify these entropy values.

3- Authors claim that this high entropy anode material goes through a conversion reaction and at the end of lithiation (at 0.01V) metallic and Li₂O phases are formed, however they have not presented a single material characterization data that shows the formation of metallic or Li₂O phases. Authors have not even discussed their CV pattern to support their argument. Previous studies on single metal anode materials, for example MnO has successfully shown the formation of Mn metal wherever it is applicable. For example J. Power Sources 196 (2011) 6802.

4- Authors write on the p13 "it can be concluded that the rock-salt structure is preserved during the cycling process and it serves as a host matrix for the redox processes. Unreacted TM-HEO can be ruled out...". Authors should explain how rock salt structure is preserved when this high entropy anode material completely converts to metallic and Li₂O phases. Which elements are still making the host rock salt structure?

5- Figure 3 shows the SAED patterns of initial, fully lithiated (at 0.01V) and delithiated electrode material. Rock salt structure related diffraction rings appear more prominent in full delithiated material compared to lithiated (after one cycle) material. How it can be justified when reaction mechanism states that during delithiation rock salt anode material converts to metals and Li₂O and then convert back to rock salt structure during delithiation. Then after 1st cycle there should be significantly more amount of rock salt phase compared to the amount of rock salt phase in the delithiated electrode and rock salt phase related rings should appear strong after one cycle.

Answers to the Reviewers: High Entropy Oxides for Reversible Energy Storage (NCOMMS-18-07227B)

We thank the reviewers for their detailed comments and valuable suggestions. In the revision process, we tried to answer all the questions and conducted additional TEM, CV, XRD and electrochemical measurements and, with that, hope to meet the expectations of the reviewers.

In this letter, we are addressing the different questions one by one. For better readability, we used black font for the comments and questions, blue font for the corresponding answers and green font for the paragraphs added to the manuscript.

Reviewer #1 (Remarks to the Authors):

The authors prepared a five element oxide crystallizing in the NaCl structure and investigated the electrochemical behavior as electrode material for Li based batteries. The samples were characterized by TEM based methods and the electrochemical behavior is compared to oxides consisting of four cations. The better electrochemical behavior of the five element material is traced back to entropy stabilization. While investigation of such oxides for LIB applications is interesting the authors do not provide much insight into the reaction mechanisms occurring during Li uptake and release. Operando XRD demonstrates that the NaCl structure is destroyed on the long range scale. SAED shows reflections of very small and may be highly defective NaCl-type crystalline regions. I'll suggest addressing the points raised below and resubmit the manuscript

Additional comment: Bragg reflections are not signals!

We apologize that we did not correct this term in the main document. Somehow we completely overlooked this comment and we noticed this after the submission. The editor, Dr. Yaoqing Zhang, kindly gave us the permission to add this comment to the already submitted version. Therefore we ensure, that we will change the word "signals" to "reflexes" after the manuscript was evaluated and accepted or denied.

Question #1:

If Li enters the NaCl structure only tetrahedral sites are available. But what is the Li-M distance? I'll assume much too short and in my opinion intercalation can be ruled out.

Statement of the authors:

We agree with the reviewer. Since this is an important point, we added some more discussion to clarify this and provide some examples.

Text added to the main manuscript:

The precondition for insertion in the TM-HEO would be that the Li ion fits into the rock-salt lattice, where all octahedral positions are occupied by cations. As per Pauling's rule, the possibility of accommodating Li⁺ in the tetrahedral position is energetically costly considering that the ratio of ionic radii (Li⁺ to O²⁻) is greater than 0.414 (the upper limit for accommodating an ion in the tetrahedral position)¹⁵. Additionally, this would lead to strong repulsive interactions between the cations in the lattice¹⁵. As known from the literature, transition metal oxides are prone to conversion reactions, i.e., during the charging-discharging cycle complete phase transformations may occur^{14,16}. Consequently,

it is reasonable to assume that reversible conversion reactions take place in TM-HEO, which either completely or partially reduce the metal ions upon lithiation.

References added:

14. Poizot *et al.*, Nature 407, 2000
15. Chiang *et al.*, Physical Ceramics, Wiley, 1997
16. Helen *et al.*, Sci. Rep. 5, 2015.

Question #2:

The authors mention that the experimental capacities match with those calculated but do not present formal reactions with accompanying capacities.

Statement of the authors:

We added a paragraph to the paper, describing several examples and explaining the capacity calculation in more detail.

Text added to the main manuscript:

For conversion type materials, the theoretical capacity is directly related to the amount of electrons transferred per formula unit (Supplementary Equation SE1). The basic reaction for a binary metal oxide can be represented as $MO_x + 2xLi \rightarrow M + xLi_2O$. Because it is not known what reactions are occurring during the redox processes in the case of TM-HEO (additional processes might be alloying of Zn with Li, formation of intermetallic phases etc.), we cannot predict with high accuracy the theoretical capacity of our compounds. However, we believe that it is in the range of capacities reported for divalent oxide conversion materials. (700 – 1000 mAh g⁻¹).

Added to the Supplementary information:

Supplementary equation SE1: Calculation of the theoretical capacity

$$Q_m = (z \cdot F) / M$$

Q_m represents the theoretical capacity, z the amount of transferred electrons, F the Faraday constant, and M is the molar mass of the active material. The specific capacity for compounds of almost the same molecular weight and redox state is very similar (e.g. $Q_m(\text{CoO}) \approx Q_m(\text{NiO}) \approx 715 \text{ mAh g}^{-1}$).

Question #3:

The authors did not mention that Zn^{2+} can be alloyed with Li often leading to higher capacity than expected for reduction of Zn^{2+} to Zn.

Statement of the authors:

We thank the reviewer for this comment and fully agree. We added a paragraph to the paper and an additional figure, which shows the CV of the different compounds. Nevertheless, the Zn^{2+} ion has an atomic ratio of only 10% compared to the whole compound. Therefore, the fraction of Zn, which could alloy would lead to an additional capacity of around 37 mAh/g. The alloying reaction between Zn and Li is described to have one e⁻ per formula unit. This gives a specific capacity of 372 mAh/g²⁹. Since we

have 10 at. % Zn in our compound, this would give us a total additional capacity of 37 mAh/g, which would be around 6% of the total capacity at C/10 (600 mAh/g). The alloying reaction between Zn and Li will take place at low potentials, around 0.2 V²⁸. The CVs (Supplementary Figure S8) show a current increase in this region, but this increase does not change when Zn is substituted from the 5-component system.

Text added to the main manuscript:

Despite potential alloying of elemental Zn with Li²⁸, we did not find any signs of this reaction. Usually, the formation of ZnLi occurs at a low potential of around 0.2 V²⁸ and would lead to an additional gain in capacity of $\sim 40 \text{ mAh g}^{-1}$ for the TM-HEO²⁹. Although the CV curves (Supplementary Figure S10) for TM-MEO(-Zn) and the Zn-containing materials do not show significant differences, alloying of Zn with Li cannot be ruled out completely, especially for TM-MEO(-Cu), where the vast majority of Li uptake occurs at low potential. However, even if this reaction takes place in the TM-HEO, the capacity gain will only account for $\sim 6.7 \%$ of the total capacity, i.e., around 40 mAh g^{-1} .

References added:

28. Mueller et al., J. Electrochem. Soc. 164, 2017

29. Bresser et al., Chem. Mater. 25, 2013

Added to the Supplementary information:

Figure S8: Cyclic voltammograms obtained on the different single phase compounds. The curves reveal the difference in electrochemical behavior of the HEO compared to MEO. For TM-MEO(-Zn), the two anodic peaks centered around 1.7 V and 2.2 V indicate formation of NiO and CoO, respectively². In the case of TM-MEO(-Cu), a distinct reduction peak at low potential appears³.

References added:

2. Wang *et al.*, *J. Power Sources* 209, 2012
3. Mueller *et al.*, *J. Electrochem. Soc.* 164, 2017

Question #4:

It is well documented that Mg^{2+} is inactive in the potential window applied here. Hence, Fig. 5 is incorrect. I'm wondering why no four element oxide was tested which does not contain Mg^{2+} .

Statement of the authors:

Unfortunately, the four elements oxides without Mg^{2+} and Ni^{2+} could not be stabilized as single-phase compounds, even at much higher temperature. This is the reason why we did not show them, since we believe that simple mixtures of different salts cannot be compared to the single-phase compounds. Additionally, we assume, that exactly this phenomenon, that we cannot stabilize the structure without Mg^{2+} , is an indication for the stabilizing role of Mg^{2+} during the redox processes of the other elements. We changed Fig. 5 (now Figure 6, since, an additional Figure was implemented in the text) accordingly to better show our assumption that Mg and Ni are the major players with regard to structural stabilization and clarified the synthetic problem in the text. Besides, we performed additional electrochemical measurements on multi-phase compounds and show the results in the supporting information together with the XRD data (Supplementary Figure S8).

Text added to the main manuscript:

Despite efforts to synthesize every possible MEO structure, both TM-MEO(-Mg) and TM-MEO(-Ni) (i.e., $(\text{Co}_{0.25}\text{Cu}_{0.25}\text{Zn}_{0.25}\text{Ni}_{0.25})\text{O}$ and $(\text{Co}_{0.25}\text{Cu}_{0.25}\text{Mg}_{0.25}\text{Zn}_{0.25})\text{O}$, respectively) could not be stabilized as single-phase compounds, even at much higher temperatures. Therefore, we refrain from including them into the comparison of the entropy-stabilized oxides, since we believe that mixtures of different compounds are not comparable to the single-phase materials. Nevertheless, the electrochemical and XRD characterization results of the multiphase compounds are depicted separately in Supplementary Figure S4.

Revised Figure 5 (Now Figure 6)

Added sentence: M in the figure stands only for the cations Co, Cu, Zn, Ni, since Mg is usually electrochemically inactive in the used potential window.

Added to the Supplementary information:

Figure S4: a) XRD patterns of multiphase compounds TM-MEO(-Ni) and TM-MEO(-Mg). These materials could not be synthesized as single phase compounds. The corresponding electrochemical characterization is shown in panels b) and c), respectively.

Question #5:

The authors used a relatively low loading of 63%. Why? Was the loading identical for the four metal oxides? What was the loading per cm^2 ?

Statement of the authors:

The loading of the electrodes was adjusted to 63 wt% of active material, since in former studies this ratio led to good results for conversion and alloying electrodes⁴. We agree that the areal capacity will suffer from this low loading, but to investigate the principle behavior of HEOs we intended to assemble well working battery cells. Nevertheless, we built a cell with a much higher loading (almost fivefold increase) and an additional one with a weight ratio of 8:1:1 (active material, binder, carbon). The corresponding data are shown in the new Figure 5b and in the supplementary information (Figure S12). The electrode with higher loading delivered only slightly lower capacities. The areal capacity after 40 cycles at a current density of 100 mA/g was 1.3 mAh/cm².

Text added to the main manuscript:

For comparison of the “high-entropy materials”, “medium-entropy materials” and “multiphase materials”, the Coulombic efficiencies over the first 50 cycles are depicted in Figure 5a. As is seen, only the “high-entropy material” shows stable behavior and can be cycled for hundreds of times. Even higher loading of TM-HEO cells reveals Coulombic efficiencies of >98.5% (Figure 5b). In addition, Figure 5 b) shows results from the respective rate performance test, where the areal capacity after 30 cycles (current density 100 mA g⁻¹) amounts to 1.3 mAh cm⁻². Despite the fact that the loading was increased by a factor of ~5 (0.5 vs 2.3 mg cm⁻²), the specific capacity decreased only slightly (see also electrochemical performance of an electrode containing 80 wt. % of TM-HEO in Supplementary Figure S12).

Figure 5: Coulombic efficiencies of the TM-MEO and TM-HEO compounds and rate test of a higher loaded TM-HEO electrode. a) Coulombic efficiency vs cycle number for all the tested electrode materials at 50 mA g⁻¹. Only TM-HEO reveals stable cycling behavior. The multiphase material without Ni exhibits comparable efficiencies, but with much lower specific capacities of around 300 mAh g⁻¹

(Supplementary Figure S4). b) Rate performance test performed on a TM-HEO cell with an overall higher loading ($2.3 \text{ mg TM-HEO cm}^{-2}$). The current densities in b) are given in units of A g^{-1} .

Added to the Supplementary information:

Figure S12: Capacity retention and Coulombic efficiency of an electrode with a relatively high TM-HEO content (TM-HEO:binder:carbon black ratio of 80:10:10). The current density applied was 200 mA g^{-1} .

References added:

4. Breitung *et al.*, *Nanoscale* 8, 2016

Question #6:

Figure 3: It would be much better showing the amount of Li consumed during discharge/charge than the time.

Statement of the authors:

We fully agree and thus revised Figure 3. Since we cannot exactly predict how many Li are reacting during the first discharge, the direct plotting of the Li uptake might be somewhat inaccurate. Therefore, we decided to explain the Li uptake in the text and correlate the measured discharge capacity to the theoretical uptake of Li per formula unit (2 Li per formula unit for a complete conversion reaction).

Text added to the main document:

The voltage profile seen for the operando XRD measurement is plotted versus the dis/charge time. The first cycle discharge capacity amounts to around 1030 mAh g⁻¹, which corresponds to an uptake of roughly 2 Li per formula unit. The initial charge capacity is lower, being 459 mAh g⁻¹.

Question #7:

It was reported in literature that Cu²⁺ is reduced to elemental Cu during discharge and Cu dendrites are formed which increase conductivity of the electrode material thus improving cycling stability. Is this a possible scenario for the Cu containing samples?

Statement of the authors:

We thank the reviewer for pointing this out. We scanned the literature and conducted some experiments to gain insight into possible Cu dendrite formation in our material. We found similar studies, which we cite in manuscript^{25,26} and added related text to the main manuscript.

Text added to the main manuscript:

Another aspect discussed in the literature is the potential improvement in conductivity when metallic species like Cu dendrites are present²⁵, which might lead to better cycling stability, too. To examine whether Cu dendrites are formed during the conversion process, EDX measurements were conducted on electrodes after 100 cycles. However, the data show no indication of any Cu aggregation in the materials (Supplementary Figure S6)²⁶.

References added:

25. Morcrette *et al.*, Nat. Mater. 2, 2003

26. Débart *et al.*, Solid State Sci. 8, 2006

Added to the Supplementary information:

Figure S6: EDX measurements performed on an electrode after 100 cycles. No Cu aggregation nor dendrite formation were observed. The green layer in a) and b) shows the Cu distribution. The homogenous coloring indicates that no dendrites have formed, which would appear as a higher local color intensity. In c) besides Cu (green), Ni (yellow) has been added to show the equal distribution of the different elements.

Question #8:

For all samples a strong capacity loss is observed in the first cycle. The authors should make a comment on this and should estimate the amount of Li lost in the first cycle. If the loss of Li cannot be avoided (only Li for SEI formation is beneficial) conversion materials are not really relevant for industrial applications.

Statement of the authors:

The capacity loss in the first cycle is an undeniable issue, which we were not able to address yet. The low Coulombic efficiency for the first cycle is well known for alloying or conversion materials like Si or Fe₂O₃ and, as the reviewer states correctly, is one of the leading reasons why those materials are currently not much addressed in industrial applications. To counter this problem, we have to understand and precisely determine the detailed mechanisms of the here presented redox reactions. At this state of the research, we are not yet able to identify and tackle the exact material specific problems, which cause the observed low efficiency in the first cycle. Therefore, at the moment we have to address these issues by conventional methods that are used frequently for other conversion and alloying materials (e.g. ball milling etc.). We are optimistic that in future system optimizations we can reduce the initial capacity loss, e.g. by particle size reduction (many for this study used particles are micron-sized) to shorten diffusion lengths, or by better choices of binder and electrolyte (no comparisons of different binder/electrolyte systems have been conducted yet). After identifying the material specific problems, we will be able to search for tailored solutions. (Nevertheless, for curiosity, we also prepared a full cell with LTO as counter electrode (1.5 x Li content) which showed stable behavior for over 20 cycles.)

Reviewer #2

This manuscript reports the application of high entropy oxide as a stable anode material in the lithium-ion battery. Authors have shown that entropy stabilization is a critical factor for designing a multicomponent electrode material. Overall findings of the manuscript related to the high entropy anode materials are interesting but parts of the manuscript, especially its electrochemical reaction mechanism does not appear to be well justified. The manuscript can be accepted for publication if authors could justify the following queries.

Question #9:

In this study $(\text{Co}_{0.2}\text{Cu}_{0.2}\text{Mg}_{0.2}\text{Ni}_{0.2}\text{Zn}_{0.2})\text{O}_2$, containing 5 metals was synthesized and it is reported that this material shows better electrochemical stability as compared to its 4 metal derivative oxides, for example $(\text{Co}_{0.25}\text{Cu}_{0.25}\text{Mg}_{0.25}\text{Ni}_{0.25})\text{O}_2$, $(\text{Co}_{0.25}\text{Mg}_{0.25}\text{Ni}_{0.25}\text{Zn})\text{O}_2$ and so on. When comparing the 5 metal high entropy oxide with 4 metal low entropy oxide why electrochemical performance of compositions like $(\text{Co}_{0.25}\text{Cu}_{0.25}\text{Ni}_{0.25}\text{Zn}_{0.25})\text{O}_2$ and $(\text{Co}_{0.25}\text{Cu}_{0.25}\text{Mg}_{0.25}\text{Zn}_{0.25})\text{O}_2$ are not compared? Authors should report the electrochemical performance of these compositions as well to further clarify the effect of entropy on the stabilization of this anode material.

Statement of the authors:

Unfortunately, MEO materials without Mg or Ni could not be prepared as single-phase compounds. This question matches with #4 above, therefore we put special attention on this question and added new electrochemical and characterization results to the paper.

Text added to the main manuscript:

Despite extensive tests performed to synthesize every possible MEO structure, TM-MEO(-Mg) and TM-MEO(-Ni) ($(\text{Co}_{0.25}\text{Cu}_{0.25}\text{Zn}_{0.25}\text{Ni}_{0.25})\text{O}$ and $(\text{Co}_{0.25}\text{Cu}_{0.25}\text{Mg}_{0.25}\text{Zn}_{0.25})\text{O}$ respectively) could not be stabilized as a single-phase compound, even at much higher temperatures. Therefore, we do not include them in the comparison of the entropy-stabilized materials, since we believe that simple mixtures of different compounds are not comparable to the single phased compounds. Nevertheless, the electrochemical data and the XRD characterization of the multiphase compounds are displayed separately in Supplementary Figure S4.

Added to the Supplementary information:

Figure S4: a) XRD measurements of the multiphase compounds TM-MEO(-Ni) and TM-MEO(-Mg) which could not be synthesized as a single phase compound. The electrochemical characterization is shown in panels b) and c).

Question #10:

Table 1 reports the entropy values of the selected compositions of anode materials but method of entropy calculation is not mentioned anywhere in the manuscript. Authors should write the method of entropy calculation or they should point some suitable reference to justify these entropy values.

Statement of the authors:

We added a paragraph to the paper, which explains the calculation of the entropy for the different compounds. The entropy is calculated based on the general equation for Gibbs entropy, which is derived from statistical thermodynamics.

Text added to the main manuscript:

The configurational entropy was calculated using the following formula²⁴:

$$S_{\text{config}} = -R \left[\left(\sum_{i=1}^N x_i \ln x_i \right)_{\text{cation-site}} + \left(\sum_{j=1}^N x_j \ln x_j \right)_{\text{anion-site}} \right]$$

where x_i and x_j represent the mole fractions of ions present in the cation- and anion-site, respectively. The contribution of the anion site is expected to have a minor influence on

S_{config}, given that only one anion is apparent. More details about the entropy calculation for the TM-HEO and one of the TM-MEO materials are given in Supplementary Equation SE2.

References added:

24. Murty *et al.*, Butterworth-Heinemann, 2014

Added to the Supplementary information:

$$S_{\text{config}} = -R \left[\left(\sum_{i=1}^N x_i \ln x_i \right)_{\text{cation-site}} + \left(\sum_{j=1}^N x_j \ln x_j \right)_{\text{anion-site}} \right]$$

Example for TM-HEO with equimolar amount of different cations (5 cations $\rightarrow x_j = 0.2$):

$$S_{\text{config}} = -R \left((0.2 \ln 0.2) + (0.2 \ln 0.2) + (0.2 \ln 0.2) + (0.2 \ln 0.2) + (0.2 \ln 0.2) \right)_{\text{cation-site}} + (1 \ln 1)_{\text{anion-site}}$$

$$S_{\text{config}} = -R(5 * (0.2 \ln 0.2)) = 1.61R$$

For the TM-MEO

$$S_{\text{config}} = -R(4 * (0.25 \ln 0.25)) = 1.39 R$$

Question #11:

Authors claim that this high entropy anode material goes through a conversion reaction and at the end of lithiation (at 0.01V) metallic and Li₂O phases are formed, however they have not presented a single material characterization data that shows the formation of metallic or Li₂O phases. Authors have not even discussed their CV pattern to support their argument. Previous studies on single metal anode materials, for example MnO has successfully shown the formation of Mn metal wherever it is applicable. For example J. Power Sources 196 (2011) 6802.

Statement of the authors:

We thank the reviewer for this comment. We performed several experiments to identify metallic phases, but unfortunately we were not able to find any. We assume that this is a direct consequence of the mechanism leading to reduction in a rock-salt type matrix. Nevertheless, we could find an indirect indication that a conversion mechanism is taking place since we were able to detect a very weak diffraction ring at 2.68 Å. This d-value could be attributed to the Li₂O phase, as depicted in Figure 4. It has to be noted that the reflex intensities using X-ray diffraction or electron diffraction are differing due to the scattering factor of the material, but the d-value itself is not changing. Additionally, we tried to identify the redox processes displayed by the different CVs. Since the CVs are showing significant differences, we added a Figure to the supplementary information (Supplementary Figure S8) showing all CVs obtained on the single-phase compounds.

Text added to main manuscript:

To support the above hypothesis, the products of the conversion reaction (Li_2O and metallic species) were analyzed via TEM. Interestingly, no metallic phases were found, which seems to be the consequence of the aforementioned reaction mechanism (i.e., reduction inside the rock-salt matrix). Although Li_2O is known to be highly sensitive to the electron beam, the presence of Li_2O could be rationalized from the samples after the first and second lithiation cycles (see SAED patterns in Figures 3 a) and 3 b)). The d-spacing of 2.68 \AA could be attributed to the (111) diffraction of Li_2O . Because Li_2O is only forming due to the conversion reaction, this can be seen as an indirect support of the proposed reaction scheme.

Added to Figure 3:

Added sentence: The presence of a very faint SAED ring in the lithiated samples in b) and c) (d-spacing 2.68 \AA) could be attributed to the (111) diffraction of Li_2O .

Regarding the CVs:

The cyclic voltammograms (CVs) are depicted for all the systems in **Supplementary Figure S8**. The CV curves look very different for the different materials, which is another clear indication that the reaction mechanism and the electrochemical behavior can be tailored by changing the elemental composition. Removal of Zn from the TM-HEO causes a completely different electrochemical behavior during the oxidation of the compound. The CVs and differential capacity plots establish that the absence of Zn leads to a two-step oxidation process, rather than a single one as for all the other samples (**Supplementary Figure S9**). These two oxidation peaks, centered around 1.7 V and 2.2 V , resemble the formation of individual NiO and CoO ^{17,27}. This suggests that, in the case of TM-MEO(-Zn), the parent rock-salt structure is apparently not regained upon Li extraction, but instead separates into different phases.

References added:

17. Ding *et al.*, Phys. Chem. Chem. Phys. 18, 2016.

27. Wang *et al.*, J. Power Sources 209, 2012

Text added to the Supplementary information:

Regarding the CVs:

Figure S10: Cyclic voltammograms obtained on the different single phase compounds. The curves reveal the difference in electrochemical behavior of the HEO compared to MEO. For TM-MEO(-Zn), the two anodic peaks centered around 1.7 V and 2.2 V indicate formation of NiO and CoO, respectively². In the case of TM-MEO(-Cu), a distinct reduction peak at low potential appears³.

References added:

2. Wang *et al.*, *J. Power Sources* 209, 2012
3. Mueller *et al.*, *J. Electrochem. Soc.* 164, 2017

Question #12:

Authors write on the p13 “it can be concluded that the rock-salt structure is preserved during the cycling process and it serves as a host matrix for the redox processes. Unreacted TM-HEO can be ruled out...”. Authors should explain how rock salt structure is preserved when this high entropy anode material completely converts to metallic and Li₂O phases. Which elements are still making the host rock salt structure?

Statement of the authors:

To be honest, we are not completely sure at this early time of investigations. We assume that the answer is related to Question #9 and the first part of Question #4, where we mention that the

compounds cannot be stabilized as single-phase materials when Mg or Ni is missing. Since Mg/MgO is electrochemically inactive in the given potential range, we believe that the MgO (rock-salt structure) is the main contributor retaining the rock-salt lattice when the other elements are reduced to their metallic state. SAED rings from electrodes, which were cycled for 10 cycles, demonstrate that a rock-salt structure is preserved, but do not indicate high crystallinity as would be expected for unreacted material. Therefore, we assume that a highly disordered rock-salt structure composed of MgO and other cations is preserved during cycling and leads to the high stability and high Coulombic efficiency. We added some text to the manuscript and we revised Figure 5 (now Figure 6) to clarify our understanding.

Text added to the main manuscript:

We believe that the instance that single-phase compounds without Mg^{2+} or Ni^{2+} apparently cannot be synthesized is an indication of the stabilizing role of those ions during the redox processes. The pivotal role of Mg^{2+} in keeping the HEO structure intact can be explained by the fact that Mg is inactive in the given potential range.

Added sentence: M in the figure stands only for the cations Co, Cu, Zn, and Ni, since Mg is electrochemically inactive in the potential range applied here.

Question #13:

Figure 3 shows the SAED patterns of initial, fully lithiated (at 0.01V) and delithiated electrode material. Rock salt structure related diffraction rings appear more prominent in full delithiated material compared to lithiated (after one cycle) material. How it can be justified when reaction mechanism states that during delithiation rock salt anode material converts to metals and Li_2O and then convert back to rock salt structure during delithiation. Then after 1st cycle there should be significantly more amount of rock salt phase compared to the amount of rock salt phase in the delithiated electrode and rock salt phase related rings should appear strong after one cycle.

Statement of the authors:

We thank the reviewer for this question, as it is indeed somewhat unclear. The SAED patterns in Figure 3 are there to indicate that the rock-salt structure is preserved. The brightness and strength of the diffraction signals varies also depending on, for both measurements, of the thickness of crystallites in the selected area contributing to diffraction. This leads to a different intensity of the diffraction rings and therefore it seems like there is more rock-salt type material in the lithiated state. We added an SAED pattern obtained on material cycled over 10 cycles still showing rock-salt structure.

Text added to the main manuscript:

Supplementary Figure S10 shows an SAED pattern obtained on an electrode after 10 full cycles, still showing the rock-salt structure.

Added to the Supplementary information:

The different intensity in the SAED rings displayed in Figure 3, come from a slightly different thickness of crystallites in the SAED area. The main intention to show these SAED rings is, to support the finding that a rock-salt structure is being preserved through the whole cycling process. Therefore Supplementary Figure S10 a) shows an SAED pattern of a 10 times cycled electrode, where it is clearly observable that the rock-salt structure can be found even after prolonged cycling. The crystallites could even be found in HR-TEM micrographs as can be seen in b).

Figure S10: a) SAED pattern of an electrode after 10 cycles. As is seen, the rock-salt structure is retained, even after prolonged cycling. The presence of crystallites is clearly evident in the HR-TEM micrograph in b).

REVIEWERS' COMMENTS:

Reviewer #1 (Remarks to the Author):

The authors properly addressed my critics and suggestions. Hence I'll suggest acceptance of the paper.

Reviewer #2 (Remarks to the Author):

I have carefully read the reviewers response to my comments on their manuscript titled "High Entropy Oxides for Reversible Energy Storage" and I believe that after revision the manuscript can be accepted for publication in Nature Communication.